



# Fluorescent double labelling of F-actin in Foraminifera: evaluation of granular pattern F-actin organisation in reticulopodia

Jan Goleń [1], Jarosław Tyszka [1], Karolina Godos [1]

[1] ING PAN – Institute of Geological Sciences, Polish Academy of Sciences, Research Centre in Kraków, Biogeosystem
Modelling Group, Senacka 1, 31-002 Kraków, Poland

*Correspondence to*: Jan Goleń (ndgolen@cyf-kr.edu.pl)

**Abstract.** This paper presents novel results of experiments focused on actin cytoskeleton organisation in pseudopodial structures of foraminifera. The main aim of this research was to test the hypotheses proposed to explain the previously reported granular pattern of F-actin labelling in several species of foraminifera with SiR-actin fluorescent probe. These hypotheses include the possibility that SiR-actin enhances the F-actin polymerisation or that it binds to other than F-actin compounds within foraminiferal cells, resulting in capturing the staining artefacts. The series of replicated experiments conducted on small miliolid *Quinqueloculina* sp. included double staining of F-actin with SiR-actin and phalloidin-based probes, complemented with observations of reticulopodia under the polarised light to identify the granules showing birefringence. All of performed experiments demonstrate highly congruent results expressed in SiR-actin and phalloidin co-labelling patterns, especially pronounced with the small granular objects (SiR-actin labelled granules or ALGs). Furthermore, the birefringence strongly tends to characterise primarily areas stained with both of the probes. These results rule out staining artefacts and support existence of actin cytoskeleton associated with micrometre-size motile granules. We discuss possible implications of this granular pattern that facilitates efficient formation and fast reorganization of extended granuloreticulopodia and other pseudopodial structures. If this is the case, they are one of the key evolutionary adaptations of these organisms that most likely predates the evolution of the foraminiferal tests that started in the early Palaeozoic.

## 1 Introduction

Pseudopodia in foraminifera have very specific appearance that distinguishes them from similar structures in other organisms. They usually form structures called either reticulopodia or granuloreticulopodia with elaborate hierarchical networks of thicker and thinner threads, which have a strong tendency to anastomose. Granuloreticulopodia contain large amount of granular objects moving bidirectionally along their threads. Granular appearance of pseudopodia is unique to foraminifera, hence it is of great taxonomic importance (Bowser and Travis 2002). Structural and physiological studies of pseudopodia in foraminifera are critical to understanding the evolution, morphogenesis, physiology, and ecology of these organisms. Lamellipodia (Travis et al. 1983) and globopodia represent additional types of pseudopodial structures, being involved in the chamber formation and calcification (Tyszka et al., 2019). In general, pseudopodia play a crucial role in motility, attachment, feeding, chemosensing, and shell formation in foraminifera (Goldstein 1999; Tyszka et al., 2019). First observation of pseudopodia and



their role in the attachment and motility of foraminifera dates back to Dujardin (1835), for the review of early observations of pseudopodia see Goldstein, 1999). The progress in their research, however, accelerated with the introduction of electron microscopy and fluorescent microscopy, which allowed for imaging of their ultra-structure in high resolution and identifying basic components of the networks.

Pioneering ultrastructural studies utilising electron microscopy by Wohlfarth-Bottermann (1961) Hedley et al. (1967), Marszalek (1969), later reviewed by Travis and Bowser (1991) demonstrated that key structural elements of pseudopodial networks in foraminifera are cytoplasmic microtubules and lipid membranes. Apart from the purely structural functions, microtubules enable intracellular transport of the vesicles and organelles, by providing the physical pathways along which the transport occurs (see Travis and Allen 1981; Travis et al. 1983; Bowser and Rieder 1985; Travis and Bowser 1986; Langer

40  1992).

The additional structural element hypothesised to be involved in formation of pseudopodia are cytoplasmic microfilaments, which consist of filamentous form of actin (F-actin). Actin is one of the main structural proteins in the eukaryotic cells and occur in two main forms: (1) G-actin, which is a globular protein and an unbound subunit of (2) F-actin – a linearly arranged polymer building the microfilaments (Straub 1942). Generally, microfilaments in eukaryotes are involved in motility as well

as control of the cell shape (Travis and Bowser 1986; Blanchoin et al. 2014; Small and Resch 2005), so their presence in structures like pseudopodia, where these processes occur with the high intensity, was expected. Early studies utilising the fluorescent microscopy provided somehow inconsistent evidence to what extend the actin filaments are present within the pseudopodia of foraminifera. Those studies were conducted using either phalloidin staining or indirect immunofluorescence on fixed specimens on various species (mostly on *Allogromia* sp. and *Reticulomyxa* sp., as well as on *Astrammina rara*).

Phalloidin staining is the classical method of imaging of the actin cytoskeleton in Eukaryotes, however, it poses significant limitations. The most important is that it is mainly suitable for the fixed samples because cell-permeability of phalloidin-based probes is low (Melak et al. 2017). In consequence, sample preparation requires chemical fixation and permeabilisation of the cell membrane. Such procedures usually disrupt native structures of the cell and may likely result in observation of artefacts (Melan 1994).

Microfilaments were first identified in foraminiferal pseudopodial network in TEM images by Schwab and Schwab-Stey (1972, 1973). Travis and Allen (1981) identified ~50Å thick filaments in the TEM images of pseudopods of *Allogromia laticollaris*. In attempt to test the possibility that these filaments are the form of actin they tried to decorate them with myosin sub-fragment 1 (myosin S1). As they reported the filaments in question did not bind myosin S1, leading them to conclusion that the nature of these structures differ from the actin. Koonce et al. (1986b) demonstrated by fluorescent imaging, using the

rhodamine-phalloidin, presence of microfilaments spread throughout all pseudopods.

It was reported that the F-actin is not uniformly distributed throughout the network (Travis and Bowser 1986; Bowser et al. 1988). Travis and Bowser (1991) stressed that actin is limited to patches that tend to cluster at the attachment sides of reticulopodia to the substratum. In the trunk reticulopodia within the proximal part of the network, fluorescent staining revealed presence of linear fibres (Bowser et al. 1988). In more distal regions it is associated primarily with regions of pseudopod-



substratum contact (actin is present only in limited number of plaques or teardrop-shaped foci). Moreover, visualisation of F-actin with phalloidin-staining and indirect immunofluorescence demonstrates similar patterns. The main difference was that the pattern shown by indirect immunofluorescence was more diffused, probably due to affinity of the antibodies to the actin subunits (Bowser et al. 1988). Since, it was hard to preserve actin cytoskeleton in *Allogromia* using protocols that included permeabilisation step before fixation, Bowser et al. (1988) concluded that the F-actin within *Allogromia* reticulopodia is labile.

The same study showed that F-actin has different distribution in pseudopodium than tubulin.

In contrast to *Allogromia*, the staining pattern of F-actin in *Reticulomyxa* resembled closely the staining pattern of tubulin (Koonce et al. 1986b). Moreover, presence of F-actin was corroborated not only by the rhodamine-phalloidin staining but also by the decoration with myosin S1. Additionally, they reported that certain areas of pseudopodia displayed birefringence, i.e. the refractive index of the material they contain depends on the polarisation of the light and on direction of its propagation.

The apparent brightness of these structures under polarisation microscopy depends on the direction of the light polarisation. This characterises the core part of the pseudopods, as well as bulbous tip of the pseudopodia and expanded cytoplasmic 'droplets' documented by Koonce et al. (1986b, fig. 3). Similar optical property of the pseudopodia of *Allogromia laticollaris* (Arnold) was shown before by Travis and Allen (1981) and Travis et al. (1983).

Novel method of F-actin staining has been recently developed enabling for live-staining experiments (Lukinavičius et al.,

2014). It has been successfully applied for studies on living foraminifera (Tyszka et al., 2019). Another feature of organisation of the actin cytoskeleton in foraminiferal pseudopods has been identified using this method. Its application of this probe demonstrated presence of unique granular actin-labelled objects that display vary rapid bidirectional movement along the pseudopodia (see Tyszka et al., 2019; Frontalini et al., 2019; Goleń et al. 2020). This actin staining pattern is unusual in comparison to most other eukaryotes and it has not been identified clearly in previous studies concerning actin cytoskeleton

in foraminifera. For that reason, it is thought that those granular structures may represent staining artefacts (Bowser 2019, Toyofuku 2019, Goleń et al. 2019a; Goleń et al. 2019b). To account for this possibility, the term SiR-actin labelled granules has been coined to describe them (Goleń et al. 2020). The presented study primarily addresses the question, whether they are experimental artefacts or they represent physiological and functional forms of F-actin in foraminifera.

## 2 Material and methods

### 2.1 Specimens

Foraminifera were cultured in Cracow Research Centre of ING PAN in 10 l and 30 l aquaria with 12-12 day-night photoperiod. The source of the samples was large Indo-Pacific Reef Aquarium in the Burgers' Zoo (Arnhem, The Netherlands). It has been reported that this tank is populated by at least 50 different species of foraminifera, including small and large benthic foraminifera (Ernst et al. 2011). The experiments were performed on small miliolids attributed to *Quinqueloculina* sp. that

represents tubothalamean foraminifera (see Pawlowski et al. 2013). This taxon presented the most stable and predictable reticulopodial activity that was suitable for replicated experiments. The intact structure of the pseudopodial networks of this



species can be easily preserved using the chemical fixation in comparison with other tested taxa, such as *Amphistegina* sp., *Ammonia* sp., *Peneroplis* sp.

## 2.2 Sample preparation

Several (c. 20-30) individuals have been collected from the aquaria using small paint brushes and/or Pasteur's pipets and transferred to Petri dishes where they were cleaned with 10/0 Kolinsky Red Sable paint brushes under the stereomicroscope Nikon SMZ1500. In this way the debris and the algae covering the tests or captured by pseudopods were gently removed. The specimens were transferred to another clean Petri dish. They were kept without feeding in darkness overnight to reduce the possible autofluorescence either from the endosymbiotic or digested algae. On the next day the individuals were checked for

the activity of pseudopodia and those who extended the pseudopodia were transferred to the ibidi® Glass Bottom Dishes (3-5 individuals were used for each dish), in which they were fixed, stained and imaged.

## 2.3 Fixation

The specimens were fixed after they re-extended the pseudopodia and attached to the bottom glass. The fixation procedure (concentration of the solution and fixation time) was optimised by trial and error method to maximally preserve the live-like

structure of the pseudopodial network (see section 1 of Supplementary Materials and MovieS1 in Video Supplement). This procedure involves preparation of the fixation solution, containing the 200 µl of glutaraldehyde for the 800 µl of artificial sea water (ASW). 1 ml of such solution is then carefully added under the fume hood to the Petri dish containing several small miliolids in 1000 µl of ASW. After 2 min. of fixation, the samples were washed 3 times with ASW and stained.

## 2.4 Staining

Two fluorescent dyes were added to the ibidi® Glass Bottom Dish immediately after fixation: SiR-actin stock solution 1-2 µl for 1000 ml, 10-20 µl of Phalloidin Atto 488 stock solution for 1000 ml of ASW. Staining was performed overnight in dark. On the next day, the specimens were washed in ASW 3 times and imaged. After staining and washing some of the samples were dried and embedded in Araldite (epoxy). This step was added optionally to prevent movement of the parts of the pseudopodia. In some cases, when the samples are placed in the water, the fluid motion causes slightly chaotic movement of

the whole pseudopodium. Since the imaging of different fluorescent channels is not simultaneous, a slight displacement of the areas stained with different probes may lead to the lack of co-localisation. However, embedding in araldite significantly decreases quality of the imaging with transmitted light (for those samples only the fluorescent channels are shown).

## 2.5 Imaging

Fluorescent, polarisation and DIC imaging was performed under the Zeiss Axio Observer Z1 inverted fluorescent microscope.

For the SiR-actin channel filter set 50 (excitation filter BP 640/30, beam splitter FT 660, emission filter BP 690/50) was used, whereas for the imaging of staining pattern of Phalloidin Atto 488 was conducted with the filter set 46 (excitation filter BP



500/20, beam splitter FT 515, emission filter BP 535/30). For each location within pseudopodium z-stacks has been taken to capture the 3D structure of pseudopodium. Overview image of some individuals were taken with Canon DS 126231 and stereomicroscope Nikon SMZ1500. If necessary, fluorescence images were processed using FIJI software to remove the
background noise. Additionally, imaging of the pseudopodium of an unstained individual with the same light source intensity and exposure time was done to control autofluorescence.

## 3 Results

### 3.3 Control for autofluorescence

Profile of the intensity of fluorescence along the line that crosses the pseudopodium shows low level of the fluorescence
intensity in the unstained (control) individual. Also in this individual the variability of the intensity is low, there is no significant difference between the pseudopodium and the background. The individual labelled with SiR-actin and Phalloidin Atto 488 displays much higher intensity levels and variation of the intensity with the intensity peaks in the same location. The relative height of the peak is larger for the SiR actin channel (Fig. S3 in Supplementary Materials).

### 3.2 Colocalisation of signal from SiR-actin and Phalloidin Atto 488 in fluorescently stained pseudopodia

The staining of reticulopodium with both of fluorescent probes was successful (Figs. 1-3). However, the same objects, which were in focus in the SiR-actin channel, appeared out of focus in the same z-position in the Phalloidin Atto 488 channel. Moreover, z-position in which given object appeared the sharpest in the Phalloidin Atto 488 was shifted away from the objective for about 620-930 nm in relation to the z-position in which the same object was in focus in the SiR-actin channel (Fig. S4 in the section 3 Supplementary Materials and Movie S2 in Video Supplement). It is possibly caused by dispersion.
The light of different wavelength is focused at different positions as refractive index of a medium depends on the wave length (Stanley 1971). To correct for this, after recording the z-stack, we checked which of the particular focus planes correspond to each other between two fluorescent channels. After correction for dispersion, the patterns of the distribution of the signal in both channels turned out to be very similar. The areas of the most intensive staining co-localised significantly. Both probes stained the most intensely the granular objects (see Fig. 1), however, the whole reticulopodium also displayed weaker
fluorescence. The edge of lamellipodia attached to the substrate showed distinct staining by both probes (Fig. 2). The relative intensity of signal in this area compared to the intensity of fluorescence throughout the whole pseudopodium appeared to be higher for Phalloidin Atto 488 than for SiR-actin.

### 3.3 Birefringence in pseudopodia

Images captured with polarised light demonstrate that certain structures within the pseudopodia contain birefringent material.
This property manifests itself as significant changes of the brightness of these structures depending on the polarisation (Fig. 3C-3D). They tend to overlap with the small granular structures that are labelled with SiR-actin and Phalloidin Atto 488 (Figs.



3E-F). The birefringent material is not restricted to the granular structures, but is also present in other areas stained with SiR-actin. This birefringence is observed both in the living individuals and in the fixed ones. Linear structures stained with the SiR-actin may correspond to actin filaments interacting with MTs.

## 4 Discussion

The fact that three independent methods indicate presence of the F-actin in the same location within the reticulopodium suggests that the granular pattern of staining of F-actin in foraminifera with SiR actin fluorescent probe is not accidental and observed granules indeed contain considerable amounts of F-actin. This leads to the question why the phalloidin stained granules have not been described in previous studies that contrasts with the SiR-actin stained granules observed recently during live experiments (Tyszka et al. 2019; Goleń et al. 2020).

There are several possible answers to this question. It is likely that standard fixation methods make ALGs very difficult to preserve during fixation. This may be due to at least two different mechanisms. Firstly, when pseudopodia are fixed with the relatively low concentrations of the glutaraldehyde and/or paraformaldehyde, the foraminifera might not be fixed immediately. Instead, for some limited time measured in seconds, such individuals might retain some of their vital functions including motility. As the higher and higher concentration of the aldehydes perfuses into the cell it becomes less and less motile. The cell at first tries to retract the pseudopodia, however, the process does not occur uniformly. Some parts of the pseudopodia are retracted faster, and in consequence some components of the pseudopodia evacuate quicker. This most likely happens to some form of actin cytoskeleton in reticulopodia. The SiR-actin labelled granules may facilitate fast and efficient retraction of actin components to the shell. Their primary role is probably bidirectional transport of prefabricated or recycled portions of actin meshworks. Hypothetically, other components, such as MTs, may be less labile and require much longer time to transport. Such components are easier to stabilise during fixation. This issue requires separate dedicated experimental studies. If it turns out to be true, it will explain why actin structures in pseudopodia of foraminifera are labile as reported by Bowser et al (1988). Another mechanism may be connected to overall tendency of pseudopodia to lose granular appearance after fixation (section 1 in Supplementary Materials and Movie S1).

Both main components of cytoskeleton, i.e. tubulin and F-actin, display the birefringence (Kakar and Bettelheim 1991; Oldenbourg et al. 1998). In pseudopodia both of them often co-occur with each other that is not by its own an unequivocal evidence to what extent F-actin is present in the pseudopodia of foraminifera. This is especially true for the core part of reticulopodia, where MT is certainly present (see Travis and Allen 1981; Travis at al. 1983). However, bulbous tips of the pseudopodia and expanded cytoplasmic 'droplets' also show birefringence (see 'droplets' in the fig. 3 in Koonce et al. 1986b), although the presence of microtubules in the external layer of 'droplets' seems unlikely, due to geometry of this structures. Thus the birefringence of this areas is most likely caused by presence of F-actin. It is important to emphasise that in spite of the fact that such structures have not been explicitly recognized in previous studies, similar structures may be identified in the published images, such as those presented by Koonce et al. (1986a in fig 3c).



Our results support existence of F-actin structures associated with micrometre-size motile granules (SiR-actin-labelled granules or ALGs) in pseudopodial structures of foraminifera observed previously (Tyszka et al. 2019; Goleń et al. 2020). We demonstrated that ALGs can be observed not only in the living specimens but also in the fixed ones. Moreover, they can be stained by two independent methods, i.e. SiR-actin (jasplakinolide-based probe), as well as with Phalloidin-based one (Figs.1-3). Furthermore, ALGs appear to contain birefringent substances (Fig. 3). These facts allow the staining artefact hypothesis to be rejected (see Bowser 2019, Toyofuku 2019, Goleń et al. 2019a; Goleń et al. 2019b for further discussion). Presence of

ALGs in two main lineages of multichambered foraminifera demonstrated by Goleń et al. (2020) suggest that they are key evolutionary adaptation that most likely predated emergence of foraminiferal tests in the early Palaeozoic. They probably facilitate efficient formation of tests and fast reorganization of pseudopodial structures in Foraminifera. In fact, exact roles they play in those processes and their ultrastructure cannot be determined without more detailed ultrastructural studies. Goleń et al. (2019a; 2019b) suggested that they may either function as a means of transport of prefabricated F-actin to the pseudopodia

(in which case the F-actin would be enclosed by lipid membranes) and/or that observed granular form of actin cytoskeleton participates in the transport of other organelles within pseudopodia.

## 5 Conclusions

Our study included two main types of the experiments (1) double staining of F-actin in foraminifera with SiR-actin and Phalloidin Atto 488, (2) observation of reticulopodia under the polarised light in order to identify the granules showing

birefringence. Presented results refute the possibility that ALGs (SiR-actin-labelled granules) are merely staining artefacts, caused by staining-induced assembly of the microfilaments in the pseudopodia of living foraminifera. The hypothesis, that the granular staining pattern in the specimens stained with SiR-actin is caused by unspecific binding to molecules other than F-actin in foraminifera, can also be ruled out. There are still several open questions remaining to be answered, concerning the exact functions of ALGs, their ultrastructure and evolutionary importance.

**Data availability**

All data supporting the results of this article are included within the article and its Supplement and Video Supplement.

**Video Supplement**

**Movie S1. Fixation of granuloreticulopodia of *Amphistegina lessonii* d'Orbigny. Granuloreticulopodia contained a large number of rapidly moving granules before the addition of glutaraldehyde (up to c. 37 s. from the beginning of the recording). The movement**

**of the granuloreticulopodia stopped after application of glutaraldehyde (c. 37-52 s. from the beginning of the recording), and granules started to gradually disappear (c. 60-320 s. from the beginning of the recording). Finally, the granuloreticulopodia were almost completely devoid of granules after c. 300 s. since application of the glutaraldehyde. Note lack of beading response during fixation. Images were obtained with Zeiss Axio Observer Z.1 with DIC optics. Scale bar 10 μm. The video was captured with the interval of 0.607 s between subsequent frames. The time elapsed since the start of recording is displayed in the upper left corner.**

**Video data can be accessed via the TIB AV-portal at https://doi.org/10.5446/51970.**



**Movie S2. Z-stack of granuloreticulopodia of *Quinqueloculina* sp. stained with SiR-actin (red) and Phalloidin Atto 488 (green) fluorescent probes. Note the shift between two fluorescent channels. Arrow indicates the position of the SiR-actin-labelled granule (ALG) stained with both fluorescent probes. The granule is in focus in the z-position 0-640 nm in the SiR-actin fluorescent channel. From the z-position 960 nm it becomes to be out of focus. For the Phalloidin Atto 480 channel the image of the same object is in focus**
**in z-position 640-1280nm. This shift applies to other areas of the image. All images were captured with Zeiss Axio Observer Z1. Scale bar 10 µm. The distance in z-axis from the first frame is indicated in the left upper corner. Video data can be accessed via the TIB AV-portal at https://doi.org/10.5446/51971.**

**Supplementary Materials**

Supplementary Materials contains additional text concerning methods of fixation of the pseudopodia and for additional figures.

**Author contributions**

JG designed and performed the research, cultured the foraminifera, analysed the data, wrote the paper, and prepared the graphics. JT supervised this research, consulted experimental designs and their results, as well as participated in manuscript preparation. KG cultured foraminifera and participated in manuscript preparation.

**Competing interests**

The authors declare that they have no conflict of interest.

**Acknowledgments**

The authors thank Samuel S. Bowser (Wadsworth Center), Jeffrey L. Travis (SUNY), and Takashi Toyofuku (JAMSTEC) for their valuable comments on methodological aspects of fluorescent labelling, as well as Sigal Abramovich (Ben Gurion University), Ulf Bickmeyer (AWI), Jelle Bijma (AWI), and Fabrizio Frontalini (University of Urbino) for long-term
cooperation. We are grateful to Max Janse and Nienke Klerks from Burgers' Zoo in Arnhem for access to living foraminifera. This research was supported by the Polish National Science Center (Grant UMO-2015/19/B/ST10/01944 to J.G. and J.T.) and (UMO-2018/29/B/ST10/01811 to K.G., grant coordinated by Grzegorz Racki, University of Silesia).

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





Figure 1: Fluorescent staining and DIC imaging of an area of granuloretiulopodium of *Quinqueloculina* sp. (a) overall image of the
fixed individual with granuloreticulopodium attached to the glass. Black rectangle indicates area imaged in detail in (b). (b) close-
up of reticulopodium. Black rectangle in (b) shows area imaged in higher magnification in (c)-(e). (c): SiR actin staining. (d):
Phalloidin Atto 488 staining. (e): detail of the reticulopodium imaged with the DIC (Nomarski contrast). (f): the colocalisation of the
staining of SiR actin (red signal) and Phalloidin Atto 488 (green signal). Yellow areas shows the highest degree of collocalisation
between two signals. Arrowheads indicate SiR-actin labelled granules (ALGs) stained by both fluorescent probes. Large arrows
indicate other areas intenslly stained with both porbes. Those areas likely contain large amount of ALGs that are too close to each
other and thus cannot bedistinguished. Image (a) was captured with camera Canon DS 126231 and stereomicroscope Nikon
SMZ1500. Images in (b)-(f) were captured with Zeiss Axio Observer Z1.





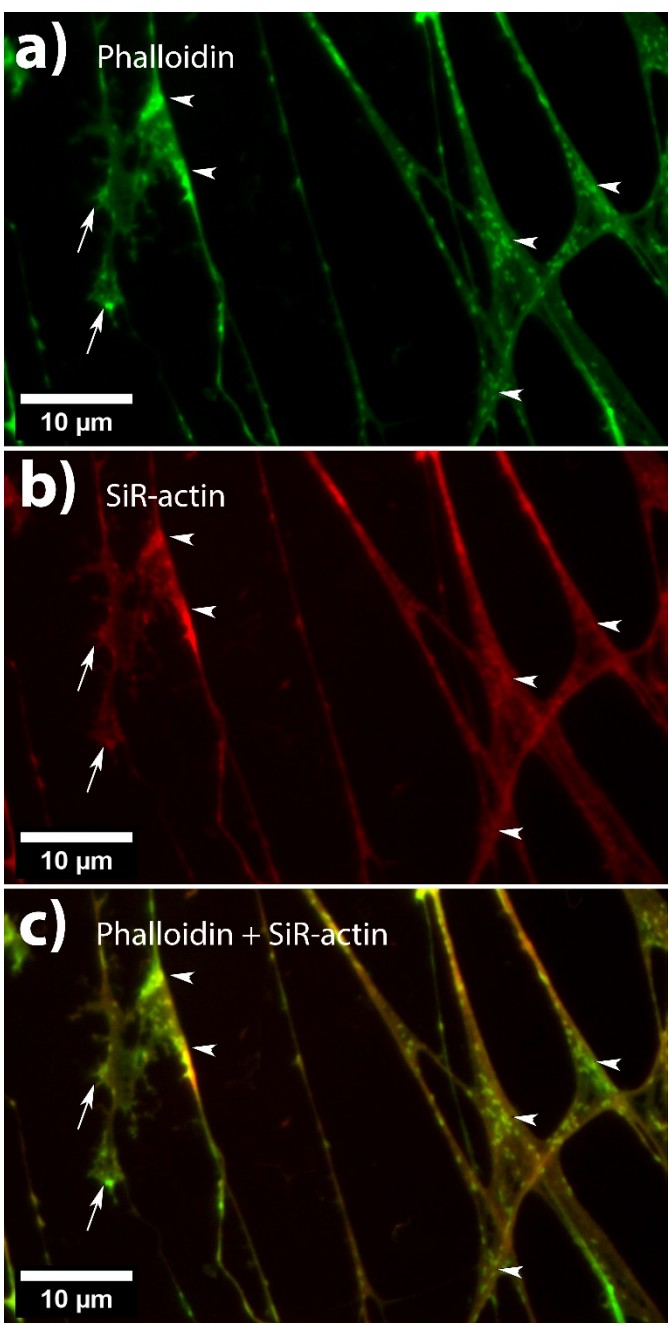

**Figure 2: Fluorescent staining imaging of an area of reticulopodium of *Quinqueloculina* sp. Arrows indicate areas of accumulation of the SiR-actin labelled granules. The arrow heads show the edge of lamellipodia attached to the glass. Scale bar equal to 10 µm. Images captured with Zeiss Axio Observer Z1.**

335



**Figure 3: Polarised and fluorescent light imaging of an area of reticulopodium of *Quinqueloculina* sp. A: the overall image of the fixed individual with the reticulopodium attached to the glass. Black rectangle in A refers to its close-up in B. The black rectangle in B indicates the postion of the area imaged in detailed in sections C-F. Scale bar in B is 20 µm. C and D: detail of the reticulopodium imaged with polarisation microsopy with light polarised with two different directions (with 75° to each other). E:SiR actin staining, F: Phalloidin Atto 488 staining. Arrows indicates the SiR-actin labelled granules (ALGs) stained with both fluorescent probes and demostrating birefringence as shown in B and C. Large arrowheads indicate attachment sites and/or areas containing large number of ALGs. Image in A was captured with camera Canon DS 126231 and stereomicroscope Nikon SMZ1500. Images in B-F were captured with Zeiss Axio Observer Z1.**