# Peer review of "Fluorescent double labelling of F-actin in Foraminifera: evaluation of granular pattern F-actin organisation in reticulopodia"

_Biogeosciences, 2021_

## Author Comment (AC4)

**Detailed response to the interactive comment by Takashi Toyofuku.**

On behalf of all authors, I would like to thank Dr Takashi Toyofuku for his valuable comments. I would like to present our detailed response to each of the referee's comments. Original referee's comments are in black, and our responses are in blue.

**General comment**

This research investigated a point that was unclear in the previously published research. Although the content is good, it is questionable whether the purpose and discussion match the common interests of the audience of this journal. In addition, I think many points can be discussed with previous studies, but they are lacking. If these points are improved, I can agree there is a great possibility that the contents of this paper can be published in Biogeosciences.

Thank you for the helpful comment. We will rewrite Introduction to show the significance of our study to Earth Sciences. We will begin with general description of foraminifera and significance, followed by general information regarding pseudopodia and their function. We will add definitions of terms and citation to the relevant literature, wherever it is necessary. As further suggested, we will add references to the literature on morphogenesis of foraminiferal tests, including research using *in silico* and *in vivo* methodologies.

Here we would like to explain important and linked issues: the taxonomy, number of specimens and SEM documentation. During preliminary stage of our experiments we used wide range of tubo- and globothalamean foraminifera (see Table R1 and Fig. R1). Tubothalamea were represented by several specimens of miliolids. By observation under the stereomicroscope, we identified two distinct types among the individuals: first one was elongated and the other is globose to ovate in overall shape of the test. We consulted the original paper that presents the diversity of foraminifera in the Burgers' Zoo marine aquaria (Ernst et al., 2011) and established that our elongate type corresponded to the individual 5 in fig. 4 in this paper. The authors identified this individual as belonging to genus *Quinqueloculina*, without specification of the species. The morphology of most of the globose and ovate individuals in our sample resembled the individual 10 in fig. 4 (identified as *Quinqueloculina bicarinata*) in Ernst et al. (2011). One individual from our individuals (F8 in Table R1) is comparable to individual 11 in fig. 4 (identified as *Miliolinella labiosa*).

Following the referees' suggestion, we conducted additional SEM imaging of specimens stored after observations. State of the preservation of the specimens was not good and a few of them were lost during the transfer to the SEM stubs. Moreover, as mentioned in the manuscript some individuals were embedded in Araldite after fluorescent dye staining (as mentioned in the manuscript). This procedure prevents from imaging them under the SEM. Three individuals we were able to document under SEM include: F1, F3 and F8. However, the specimen F3 was significantly damaged and last two chambers were destroyed. Further consultation with relevant literature allows for conclusion that the elongated individuals likely belong to *Quinqueloculina vandiemeniensis* (Loeblich and Tappan, 1994). Globothalamea were represented by a single specimen of *Heterostegina depressa* and 3 specimens of *Amphistegina lessonii*.

We decided to include into the main manuscript only those well preserved and labelled individuals with intact granular structures observed within reticulopodia. We avoided individuals presenting the beading response after fixation and/or lacking well preserved granules in pseudopodia (see Table R1). We also excluded the individuals associated with foreign objects, displaying strong fluorescence in each channel (see individuals F3, F8, F11 and F13 – Table R1). Moreover, colocalisation of the fluorescence signal is moderate or strong in all specimens that show well-preserved overall structures of pseudopodia. Even in the absence of the granules the fluorescent signal from SIR actin largely overlaps with the signal form Phalloidin Atto 488 in the actin meshwork. Only within the individuals that show beading response after fixation the colocalisation was significantly weaker. So far, we cannot find compelling explanation for this phenomenon.

We would like to emphasize that the colocalisation between signals from two probes spans across entire granuloreticulopodial network and is not limited to small restricted areas. In fact, all of the areas of the network may be viewed as a separate test of the colocalisation hypothesis.

We agree that proper taxonomic attribution is in principal an important issue that facilitates further independent replication of such experiments. However, limited taxonomic identification of the specimens does not interfere with the presented results. We tested the hypothesis pertaining to all foraminifera that present SiR-actin-labelled granules in their pseudopodial structures, Therefore, testing this hypothesis is not species specific. In light of the additional images presented in the Fig. 1 in this response, we can conclude that our results could be extended to other foraminiferal taxa.

Nevertheless, we have done our best to specify our taxonomic identifications based on available literature. Therefore, the elongated individuals are assigned to *Quinqueloculina vandiemeniensis* Loeblich & Tappan, 1994 (see Fig. R2). This miliolid species presented best labelling results (see Table R1, specimens F1, F2; compare other individuals in Fig. R1). Additional taxa included *Miliolinella labiosa* (d'Orbigny, 1839), *Heterostegina depressa* d'Orbigny (1826), and *Amphistegina lessonii* d'Orbigny (1826).

| 0          | Tanana                 | Deservices | Descention     | O a la se a lise a ti a re | December 1 in   | A .1.1242       |
|------------|------------------------|------------|----------------|----------------------------|-----------------|-----------------|
| Speci      | laxonomic              | Beading    | Preservation   | Colocalisation             | Presented in    | Additional      |
| men        | identification         | response   | of granules    | between SiR-actin          | the             | information     |
| No.        |                        | after      | after fixation | and Phalloidin             | manuscript      |                 |
|            |                        | fixation   |                | Atto 488                   |                 |                 |
| F1         | Quinqueloculina sp.,   | no         | dood           | strong                     | Figs 1 and 3 in | SEM image       |
|            | cf                     |            | 3              |                            | the             |                 |
|            | 0 vandiemeniensis      |            |                |                            | manuscript      |                 |
| F 2 | Q. Vandiememensis      | 20         | moderate       | atrona                     | Fig. 2 in the   | Embaddad in     |
| FZ         | Quinqueloculina sp.,   | no         | moderate       | strong                     | Fig. 2 in the   |                 |
|            | CT.                    |            |                |                            | manuscript      | Araldite        |
|            | Q. vandiemeniensis     |            |                |                            |                 | (epoxy).        |
| F3         | Quinqueloculina sp.,   | some       | moderate       | moderate                   |                 | Some foreign    |
|            | likely Quingueloculina |            |                |                            |                 | objects         |
|            | vandiemeniensis        |            |                |                            |                 | stained with    |
|            |                        |            |                |                            |                 | SiR-actin       |
|            |                        |            |                |                            |                 | procont         |
|            |                        |            |                |                            |                 | present,        |
|            |                        |            |                |                            |                 | SEW Image of    |
|            |                        |            |                |                            |                 | crushed         |
|            |                        |            |                |                            |                 | indvidual       |
| F4         | Quinqueloculina sp.,   | no         | weak           | strong                     |                 |                 |
|            | likely Quinqueloculina |            |                | _                          |                 |                 |
|            | vandiemeniensis        |            |                |                            |                 |                 |
| F5         | Quinqueloculina sp     | Ves        | moderate       | moderate                   |                 |                 |
| 10         | likely Quinqueleeuline | yes        | moderate       | moderate                   |                 |                 |
|            | likely Quinqueloculina |            |                |                            |                 |                 |
|            | Dicarinata             |            |                |                            |                 |                 |
| F6         | Quinqueloculina sp.,   | no         | weak to        | moderate to strong         |                 | Embedded in     |
|            | cf.                    |            | moderate       |                            |                 | Araldite        |
|            | Q. vandiemeniensis     |            |                |                            |                 | (epoxy).        |
| F7         | Quinqueloculina sp.,   | ves        | weak           | weak to moderate           |                 |                 |
|            | likely Quinqueloculina | · ·        |                |                            |                 |                 |
|            | hicarinata             |            |                |                            |                 |                 |
| EO         | Miliolinollo lobicco   | 20         | wook to        | modorato                   |                 | Somo foroign    |
| го         | WIIIOIIITEIla labiosa  | no         | weak lo        | moderate                   |                 | Some loreign    |
|            |                        |            | moderate       |                            |                 | objects         |
|            |                        |            |                |                            |                 | stained with    |
|            |                        |            |                |                            |                 | SiR-actin       |
|            |                        |            |                |                            |                 | present, SEM    |
|            |                        |            |                |                            |                 | image           |
| F9         | Quinqueloculina sp.    | some       | weak           | week                       |                 |                 |
|            | likely Quinqueloculina |            |                |                            |                 |                 |
|            | hicarinata             |            |                |                            |                 |                 |
| E10        | Dicalifiata            |            |                | a hua ya ay                |                 |                 |
| F10        | neterostegina          | no         | weak           | strong                     |                 |                 |
|            | aepressa               |            |                |                            |                 |                 |
| F11        | Amphistegina lessonii  | some       | moderate       | moderate                   |                 | Some foreign    |
|            |                        |            |                |                            |                 | objects stained |
|            |                        |            |                |                            |                 | with SiR-actin  |
|            |                        |            |                |                            |                 | present         |
| F12        | Amphistegina Jessonii  | no         | weak           | moderate                   |                 | p. 500m         |
| E12        |                        | 10         | moderate       | moderate                   |                 | Somo foreign    |
| г 13       | Amphistegina lessonii  | some       | moderate       | moderate                   |                 | Some loreign    |
|            |                        |            |                |                            |                 | objects stained |
|            |                        |            |                |                            |                 | with SiR-actin  |
|            |                        |            |                |                            |                 | present and in  |
|            |                        |            |                |                            |                 | the Phallodin   |
|            |                        |            |                |                            |                 | Atto 488        |
|            |                        |            |                |                            |                 |                 |

**Table R1** Information regarding the individuals used in the preliminary stage of the study. The level of colocalisation was evaluated by analysing the overlay of the fluorescent images in SiR-actin and Phalllodidin Atto 488 channels (see Fig. R1). Areas that appear yellow in the overlay image indicate higher levels of colocalisation. We excluded form analyses any fluorescent objects outside the pseudopodia.